# 3D Multi-Branched SnO_2_ Semiconductor Nanostructures as Optical Waveguides

**DOI:** 10.3390/ma12193148

**Published:** 2019-09-26

**Authors:** Francesco Rossella, Vittorio Bellani, Matteo Tommasini, Ugo Gianazza, Elisabetta Comini, Caterina Soldano

**Affiliations:** 1Dipartimento di Fisica, Università di Pavia and INFN, Via Bassi 6, 27100 Pavia, Italy; vittorio.bellani@unipv.it; 2Dipartimento di Chimica, Materiali e Ingegneria Chimica “G. Natta”, Politecnico di Milano, Piazza Leonardo da Vinci, 32, 20133 Milano, Italy; matteo.tommasini@polimi.it; 3Dipartimento di Matematica “F. Casorati”, Università di Pavia, Via Ferrata 1, 27100 Pavia, Italy; gianazza@imati.cnr.it; 4Dipartimento di Ingegneria dell’Informazione, Università di Brescia, via Branze 38, 25131 Brescia, Italy; elisabetta.comini@unibs.it

**Keywords:** nano-optics, light scattering, nanowires, 3D multi-branched nanostructures, waveguiding effect in nanostructures, tin oxide nanostructure, SnO_2_

## Abstract

Nanostructures with complex geometry have gathered interest recently due to some unusual and exotic properties associated with both their shape and material. 3D multi-branched SnO_2_ one-dimensional nanostructrures, characterized by a “node”—i.e., the location where two or more branches originate, are the ideal platform to distribute signals of different natures. In this work, we study how this particular geometrical configuration affects light propagation when a light source (i.e., laser) is focused onto it. Combining scanning electron microscopy (SEM) and optical analysis along with Raman and Rayleigh scattering upon illumination, we were able to understand, in more detail, the mechanism behind the light-coupling occurring at the node. Our experimental findings show that multi-branched semiconductor 1D structures have great potential as optically active nanostructures with waveguiding properties, thus paving the way for their application as novel building blocks for optical communication networks.

## 1. Introduction

Mesoscale and nanoscale systems with a topology characterized by bends or crossings, such as *V*-, *T*- or *Y*-shaped crosswise or multi-armed structures, provide a fascinating playground for the study of transport and interference phenomena. In recent years, multi-terminal electronic devices based on semiconductors or 2D materials allowed the study of phase-coherent quantum transport, [1,2,3] as well as tunneling [4] and broken symmetry effects [5]. More recently, the formation of nanostructures with controlled size and morphology has been the focus of intensive and multidisciplinary research [6,7,8]. Such nanostructures are important in the development of nanoscale devices, and in the exploitation of the properties of nanomaterials [9]. In particular, quasi-1D metal-oxides represent a novel class of nanomaterials with increasing interest due to their functional properties and their corresponding applications in different fields, spanning from energy conversion and harvesting (thermoelectrics, photovoltaic and solar cells) [10], to gas sensing and light emission [11]. In this scenario, branched nanostructures represent unique, 3D building blocks for the “bottom-up” approach to nanoscale science and technology [12,13,14,15], regarded as very promising for several applications such as photocatalysis [16], batteries [17], gas sensing [11] and more. Recent studies have also shown that modifying the shape of the waveguide [18,19] or its surface [20] can lead to both uni- and bi-directional coupling as a result of spin-orbit interaction. More recently, tin oxide (SnO_2_) nanowires (NWs) are emerging for their versatility and potentialities: (i) organized in microarray or aligned in membranes, they can act as innovative percolating sensing elements [21], or provide the active materials for electronic channels implementing logic devices on deformable non planar substrates [22] and (ii) individual SnO_2_ NWs have been proposed as building blocks for several innovative applications such as in situ observation of electrochemical processes [23] or development of high-sensitivity humidity sensors [24]. 

In this work, we have developed multi-branched SnO_2_ one-dimensional nanostructures, characterized by a “node”— *i.e.*, the location where two or more branches originate—and we have studied how this particular morphological configuration affects light propagation when a light source (i.e., laser) is focused onto the nanostructure. 

Combining scanning electron microscopy (SEM), optical analysis along with Raman and Rayleigh scattering, we were able to further understand some of the mechanisms behind the light-coupling occurring at the node. Our experimental findings demonstrate that multi-branched semiconductor 1D structures have great potential as optically active nanostructures with waveguiding properties, thus paving the way for their application as novel building blocks for optical communication networks. 

## 2. Materials and Methods

The fabrication of SnO_2_ NWs has been presented elsewhere [25,26]. SnO_2_ NWs were dispersed in isopropanol solution and then deposited onto SiO_2_/Si substrates by spin coating, reaching an approximate density lower than 0.01 tube/μm^2^. 

Light scattering measurements were carried out in backscattering geometry, with the incident laser polarized parallel to the axis of the NWs, and with unpolarized collected light. We used a micro-Raman set-up (Horiba Jobin-Yvon Labram HR) with ~1 cm^−1^ spectral resolution, equipped with an X100 objective (laser spot < 1 μm) and an excitation laser at 632.8 nm wavelength. 

## 3. Results and Discussion

Figure 1 shows scanning electron microscopy (SEM) images of SnO_2_ NWs. NWs are dispersed from an IPA-based solution onto SiO_2_/Si substrates by the spin-coating technique; we observed numerous multi-branched nanostructures, characterized by several different geometries, lengths (several microns) and diameters (100 nm–1 μm). The details on the synthesis of these nanostructures are given elsewhere [25,26]. We focus our attention on the site where the branches originate (or equivalently merge), hereafter called the “node”. Our interest on this special location arises from the fundamental and broad significance of the concept of the “node”: dealing with networks or devices, a node is the site through which a signal can propagate and be redistributed, independently of its physical nature (e.g., electromagnetic wave, electron/spin/heat current, information, etc.). We believe, however, that this concept is multi-faceted. From a different perspective, a node can also be regarded as the site for the injection of a signal (e.g., light, current, heat, sound, etc.) which might be transferred and/or shared with the surrounding network. 

In this work, we want to demonstrate how light propagates and can be manipulated through a 3D multi-branched semiconductor nanostructure. Figure 2 shows how, using a micro-Raman objective, it is possible to focus the laser beam (He-Ne, λ ≈ 632 nm) directly on the nanostructure, in particular close to (a) the center of straight NW and (b) to a primary node (dashed square in Figure 2b). In a straight NW, we do not observe any light propagation along the nanostructure, independently of the incident light intensity (top and bottom panels of Figure 2a refer to different incident light power, as indicated). However, in the case of multi-branched nanostructure (Figure 2b), we detect light spots at the termination of each branch or in correspondence to the secondary nodes (dashed circles in main optical image). This effect is enhanced if we increase (by an order of magnitude) the power of the incident laser beam from 0.1 μW (top panels) to 1 μW (bottom panels), which leads to a clear brightness increase. We note here that, for both incident powers at the node location, output signal at the termination of each branch shows variations in intensity. Although we are currently not able to measure each individual output signal, we believe this is mainly due to the different shape, diameter, length and tip configuration of each branch [27,28]. Furthermore, if we illuminate the same nanostructure, at a location along the straight segment of one NW arm, this effect disappears (not shown). This is consistent with the observation of Figure 2a: light injection is completely absent in the straight NW of Figure 2a, even if we increase the incident light power to 10 μW. These findings clearly suggest that the node, and its illumination, play a key role as a “coupler” of the impinging laser to the different branches; that all act as nanoscale waveguides. In fact, we have observed that light is confined and propagates through each and every branch, whereas at each end the confinement vanishes and the light is emitted with a solid angle, which depends on the morphology of the branch termination. We highlight here that in our experimental configuration, the laser is focused perpendicularly to the sample, and we observe that the nodes are the only locations where the coupling of the laser into the branches is effective. This waveguiding effect is very promising for applications in nano-optics; it is relatively simple and it does not require special conditions for coupling. Due to the principle of optical reversibility [29], we expect that SnO_2_ nanostructures also behave as classical waveguides; when laser light is injected into the termination of any branch (or into the one of a straight NW), it should propagate onto the node (or the other end of the straight NW). Optimal coupling conditions would require the light to enter coaxially into an arm and be focused exactly at the tip. In this sense, the optical coupling in such conditions is quite critical and might require complex laser-nanostructure geometry, which is currently beyond our experimental capability. In addition, the morphology of the tip termination of each branch might be difficult to control, since producing smooth and flat tips is currently a challenging task.

We note here that this waveguide effect in multi-branched SnO_2_ nanostructures has been observed at nodes with dimensions comparable to the wavelength (λ) of the incident laser (λ ≈ 632 nm). In general, the transmission spectra of a multi-branched NW will depend on several parameters, such as the optical properties of the material, shape and size of the nanostructure, tip geometry and shape, and more. To envision and develop specific applications, it might be desirable to investigate the complete spectral response as a function of key parameters, e.g., by injecting light in the nodes with a tunable laser or a white light source coupled to a monocromator, and positioning a suitable detector in the neighborhood of the end tip. Although the comprehensive study of the mechanism underlying the coupling between the laser and the node is beyond the scope of this work, we investigated light scattering effects in such nanostructures to better understand the role played by the node and the branches.

Figure 3a shows the Raman spectrum of an individual straight SnO_2_ NW dispersed on a SiO_2_/Si substrate. The Raman spectrum was measured at room temperature with the laser beam focused approximately at the center of the NW. We clearly observed the lines of the tetragonal rutile structure of the SnO_2_ crystals, corresponding to the active modes E_g_ (~476 cm^−1^), A_1g_ (~633 cm^−1^) and B_2g_ (~774 cm^−1^). These Raman lines were first detected in bulk SnO_2_ crystals [30] and more recently also in straight SnO_2_ NWs with diameters in the range of 4–80 nm and lengths of up to several hundred µm [31]. In addition, we observed a strong band at ~300 cm^−1^ and a weak satellite band at ~690 cm^−1^, both of them already reported in the literature [32]. The band at ~300 cm^−1^ is theoretically predicted by a rigid ion model of the phonon dispersion curves, while the second band at ~690 cm^−1^ can be interpreted as an effect due to both confinement and disorder that can induce the LO A_2u_ mode at 687 cm^−1^ to acquire Raman activity by symmetry breaking [33].

First, we investigated the diameter-dependence of the Raman spectrum of individual and isolated straight SnO_2_ NWs. The spectra measured in NWs with diameters ranging from 250 nm to 1.1 µm show the progressive enhancement of the A_1g_ band at 633 cm^−1^, for which a Lorentzian fit allows it to obtain the corresponding intensity for each diameter, as reported in Figure 3b. While increasing the diameter, the intensity of the A_1g_ band exhibits a monotonous increase of almost two orders of magnitude. We note that, in this experiment, the incident laser polarization is parallel to the NW axis, but we are not selective in the polarization of the collected scattered light. Thus, the obtained Raman spectra were not sensitive to possible effects dependent on the diameter of the NW (optical resonances), whose detection would also be required to be selective in the polarization of the measured light [34]. Moreover, the increase of the A_1g_ band intensity can be directly related to the increase of the volume *V(r)* of Raman active material probed by the laser beam, being *r* the radius of the NW. In a simplified view, *V(r)* is the closed volume resulting from the orthogonal intersection between the laser beam (i.e., a cylinder with diameter *D* = 2R ≈ 1 µm, being R the radius of the laser beam) and the underlying NW (i.e., a uniform cylinder with diameter *d* = 2r varying between 250 nm and 1.1 µm), as schematically illustrated in the inset of Figure 3c. Although *V(r)* cannot be analytically solved, it can be numerically calculated (see Appendix A) for given values of the parameter *r*/*R* in the range between 0 and 1, as shown in Figure 3c. Overall, the *V(r)* dependence from *r/R* reproduces quite well the observed diameter-dependence of the intensity and area of the A_1g_ band, both qualitatively and quantitatively (about two order of magnitude increase). 

After studying the Raman signature of an individual SnO_2_ NW as a function of its diameter, we have then focused on how the presence of a node may affect such experimental signal and its measurement. The top panel of Figure 4 shows the Raman spectra characteristics of SnO_2_ NWs, where the laser beam has been directed towards (−) the node or (−) one of its branches, as labeled and shown in the corresponding SEM image. When the node was optically excited, we observed a clear overall signal enhancement of all the characteristic Raman bands of SnO_2_.

Furthermore, Figure 4b (bottom panel) similarly reports the Rayleigh spectra collected at the node and arm as a function of incident laser power (0.1, 1 and 10 μW), again demonstrating that the elastic scattering is strongly enhanced (approximately two-fold) at the node site. Key parameters from Figure 4 are summarized in Table 1.

We attributed the intensity enhancement observed in the Raman peaks of straight wires with increasing diameter to a “mass” effect (i.e., more material is optically excited by the beam for increasing diameter). In the case of a node, this effect is most likely to be playing an important role; in fact, the interaction volume increases where two or more branches merge together, leading to the enhancement of scattered light (see Appendix A). Nevertheless, we should also consider that different optical effects can further contribute to the light scattered enhancement. A node generally introduces changes in the system topology, morphology and structure, with an overall increase of surface roughness, defects and disorder, with a consequent increase of light diffusion. In particular, we can safely assume a node to present multiple facets which most likely can be expected to act as prisms and deflect the light within the node. Such a geminates-like feature is also likely to affect the surface morphology, perhaps by introducing additional roughness and decreasing the optical reflectivity. To this respect, we note here that due to the backscattering configuration, the measured scattered light is only a small fraction of the total one emitted on the 4π solid angle, and the Rayleigh signal (but not the Raman one) includes the back-reflected light. On the other hand, when optically exciting a node, the surface of which is quite irregular, the back-reflected light is considerably smaller, explaining why the Rayleigh signal is less enhanced than the Raman one in this particular configuration. 

Based on our experimental findings, we try to elucidate in more detail the possible mechanism behind the enhancement of scattered light. When the incident laser light enters the node, it is scattered mainly uniformly in all directions and thus enters the arms. The scattered light incident on the SnO_2_/air interface with an angle *θ* smaller that the Brewster angle *θ_B_*, crosses through the interface and is lost. Instead, the scattered light incident with an angle *θ* > *θ_B_* undergoes total reflection and is guided along the branch and re-emerges at the end of it, giving rise to the brightening spots visible in Figure 3b,c. We can tentatively estimate the value of the Brewster angle *θ_B_* = arctan*(n_2_/n_1_)*, using values for the refractive index *n_1_* of the SnO_2_ NW and *n_2_* of the contrast medium, as available in the literature. We can assume the value *n_1_* ≈ 2 characteristic of the rutile structure of SnO_2_ [35]. Besides, we can consider that the contrast medium is air (*n_2_* = 1), or eventually the SiO_2_ substrate on which the NW is deposited. In the latter case we can use a value *n_1_* ≈ 1.5 which corresponds to the average of the refractive indexes measured for different polymorphisms of SiO_2_ (corresponding to different mass density). Thus we estimate *θ_B_* ≈ 63° (air contrast medium) or *θ_B_* ≈ 53° (SiO_2_ contrast medium).

## 4. Conclusions

In conclusion, using a laser coupled to a microscope stage with sub-micron spatial resolution and laser spot size, we have shown that focusing the laser on the “node” of a multi-branched SnO_2_ nanostructure clearly leads to a robust light-guiding effect between the nodes and the end tips of the different NWs. To unveil this phenomenon, we investigated the light scattering response of a set of NWs with different shapes (straight and multi-armed) and dimensions. This allowed us to suggest a scenario where the nodes act as couplers of the laser light into the different arms, inside which the light keeps confined and propagates via total reflection below the Brewster angle. Our observation of the light-guiding effect in multi-armed SnO_2_ NWs paves the way for potential applications of these systems as building blocks for nano-optics systems and allows envisioning quasi branched metal-oxides NWs as a novel class of nanomaterials for optical communications. In particular, 3D multi-branched metal oxide structures might play an important role, both in terms of shape and materials. For example in photovoltaic conversion of photons into electricity, where branches could collect light, while transferring charges to the main nanostructures via a leaf-to-branch heterostructure [36]. Similarly, one can envision multi-materials multi-branched heterostructures where incident light upon a node, while propagating, might optically excite nearby *p-n* junction while leading to the creation of a dislocated nanoscale LED [37].

## Figures and Tables

**Figure 1 materials-12-03148-f001:**
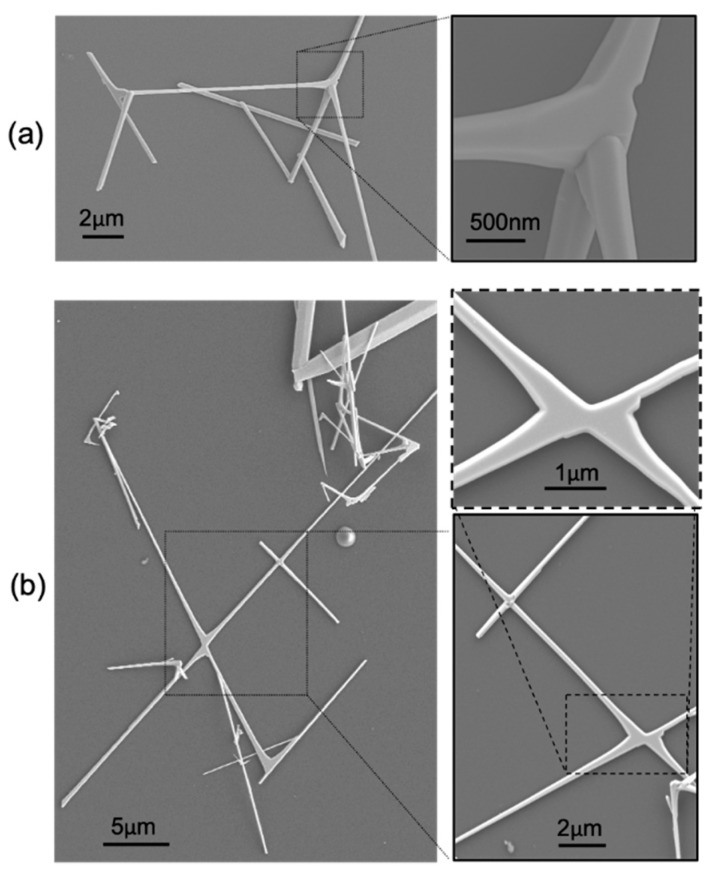
(**a**,**b**) Scanning electron microscopy (SEM) images of multi-branched SnO_2_ NWs. Insets provide detailed view of the “node”, where the two or more branches originate.

**Figure 2 materials-12-03148-f002:**
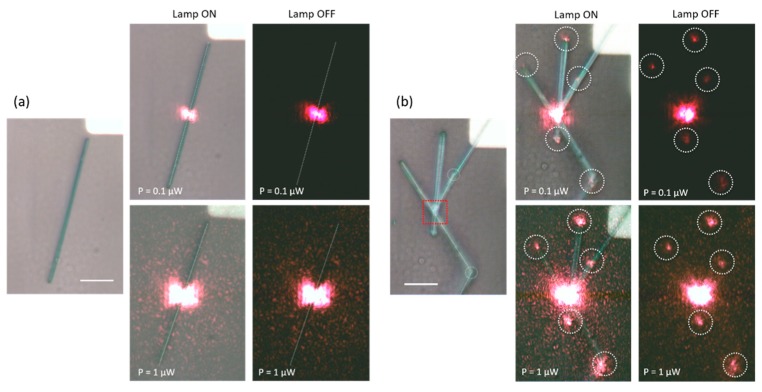
Light propagation in an individual (**a**) straight and (**b**) multi-branched SnO_2_ NWs where the laser beam has been directed to the center of the NW and to a node, respectively. For each panel, bright (lamp ON) and dark (lamp OFF) images are shown, as functions of the incident laser power (as indicated). Light does not propagate in the straight NW illuminated in its center (**a**), whereas by illuminating a node (**b**), we observed bright spots at the end of each NW branch. This effect is amplified with increasing incident light intensity, thus showing waveguide effects. The scale bar is 5 μm.

**Figure 3 materials-12-03148-f003:**
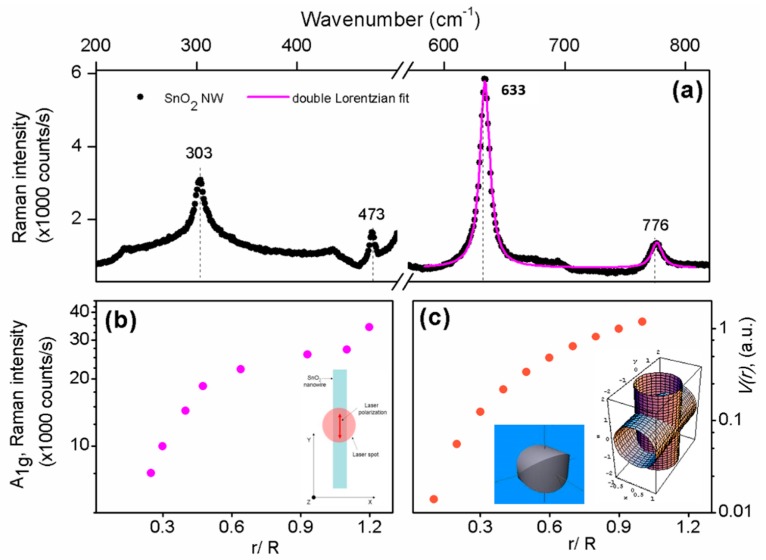
(**a**) Room temperature Raman spectrum of an individual and isolated straight SnO_2_ NW, measured at λ = 632.8 nm with the laser spot focused on the central region of the NW. (**b**) Intensity of the A_1g_ Raman line of SnO_2_ (obtained from Lorentzian fitting of the Raman band at 633 cm^−1^) as a function of the r/R ratio between the radius of the NW (r) and the radius of the laser beam (R). (**c**) Calculated Raman active volume V(r) as a function of the r/R ratio. See Appendix A for details on the calculation of the Raman active volume.

**Figure 4 materials-12-03148-f004:**
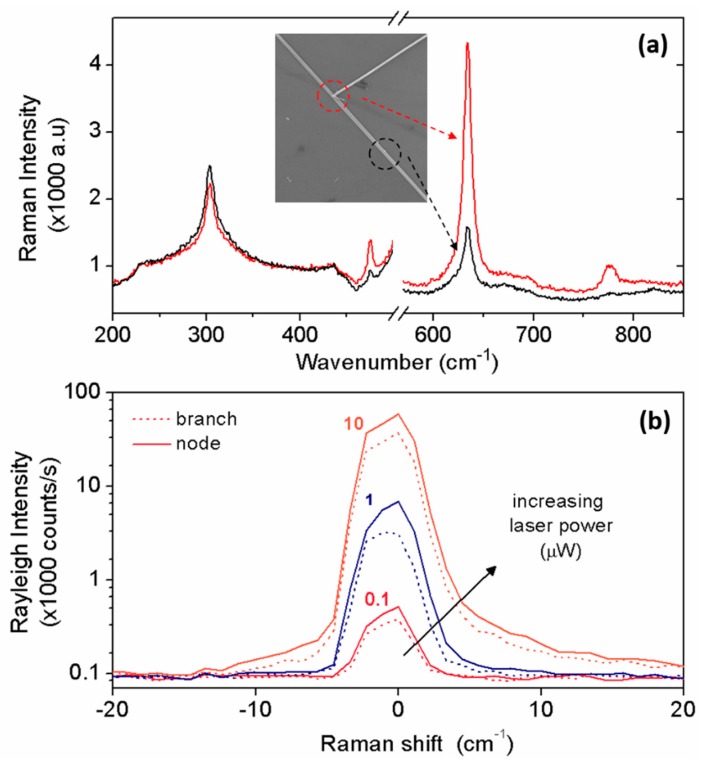
(**a**) Raman spectra measured at different locations (node and branch) of a T-branched SnO_2_ NW, as shown in the SEM image. (**b**) Rayleigh spectra measured at node (solid line) and arm (dashed line) locations for different and increasing incident laser power (0.1, 1 and 10 μW).

**Table 1 materials-12-03148-t001:** Key parameters measured from Raman and Rayleigh scattering, depending on the site of the optical excitation: branch (B) or node (N), and ratio N/B of corresponding signal amplitudes.

		Branch (B)	Node (N)	Ratio (N/B)
Raman	E_g_ (476 cm^−1^)	833	1396	1.67
	A_1g_ (633 cm^−1^)	1378	4342	3.15
	B_2g_ (774 cm^−1^)	611	1017	1.67
Rayleigh	0.1 μW	384	520	1.35
	1 μW	3181	6836	2.15
	10 μW	37,819	59,448	1.57

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
