# Peer review of "3D Multi-Branched SnO_2_ Semiconductor Nanostructures as Optical Waveguides"

_materials, 2019, doi:10.3390/ma12193148_

Round 1

Reviewer 1 Report

The authors of the article with title “3D Multi-branched Semiconductor Nanostructures as Optical Waveguides” study of the light propagation in the SnO2 nanostructures using a micro-Raman objective, and focusing the laser beam spot to the multi-branched nanostructures and observation light spots at the termination only when the light is focused to the node of each nanostructure.

The study is written using simple and understandable style, the importance of the idea of coupling the light using nanostructures is valid. However, the title of this work is about the general 3D multi-branched semiconductor nanostructures, but is using only SnO2 nanowires. Because of not presenting the other semiconductor materials in text with similar properties and effects, the authors probably can believes first that observed from them effects are valid only for the SnO2 nanostructures first. Here probably in the title of the work is needed to add “SnO2 nanostructures”, and in text the talk must be connected only with multi‐armed SnO2 nanowires, not in general.

Question: It was written: “The fabrication of SnO2 nanowires has been presented elsewhere”. Please cite the source where the preparation technology, recipe and conditions used in this study for preparation of SnO2 nanostructures were presented.

Author Response

please, see attached pdf file

Reviewer 2 Report

In this manuscript, the authors have investigated a particular geometrical configuration, i.e., “node” affects light propagation when a light source (i.e. laser) is focused onto it. Although some experimental characterizations have been made to achieve their goal, there some import concern need to be solved before it can be accepted by this journal.

In the abstract, the authors mentioned that “we were able to unveil and further elucidate in details the mechanism behind the light‐coupling occurring at the node”. However, in the main context, the authors emphasize that “Although the comprehensive and detailed study of the mechanism underlying the coupling between the laser and the node is beyond the scope of the present manuscript and unfortunately beyond our current experimental capabilities”. It seems to be a paradox between them. How about the coupling efficiency at the node and the operation bandwidth of the “waveguiding effect”? It is suggested that some discussions are made about the factors that influence the split ratio of different branches and how the split ratio will be changed with these factors (e.g., the branch width, orientation angle with the main waveguide, and etc.)? How the substrate and the polarization of incidence will affect the “waveguide effect” and the slit ratio of the “node”? More details about how to simulate the Raman scattering spectra are encouraged to be presented. It is well known that a direct waveguide cannot be excited by a normal incidence, excepting there are some in-coupling structures, e.g., coupling The fundamental reason is that the wavefront supported by the waveguide and the free pace is not matched. Therefore, it seems that the “waveguiding effect” can be easily realized by introducing scatters in the waveguide. For example, a bent waveguide has been reported can support the directional “waveguiding effect” under the normal incidence of circularly polarized light (see IEEE Photonics Technology Letters 31, 415–418 (2019)). In addition, some relevant works that utilizing the scattering or resonant “node” to directionally coupling the incidence into the waveguide along different directions have been reported elsewhere, for example, Opt. Express 19, 13831–13838 (2011),Science 346, 67 (2014), Phys. Rev. Lett. 117, 166803 (2016),and IEEE J. of Sel. Top. Quantum Electron. 24, 4700107 (2018). These references may help enrich the introduction of this manuscript.

Author Response

please see attached pdf file

Round 2

Reviewer 1 Report

The authors rewrite the text according to the recommendations. Before to publish, it must corrected some additional mistakes:

In "Result & Discussion" is written: The details on the synthesis of these nanostructures, are given elsewhere [25, 26], but in "Materials & Methods" is written: The fabrication of SnO2 NWs has been presented elsewhere [22, 23]. In Page 2 in 8 row from bottom is written: "1 um" which probably must be corrected.

Author Response

We thank you the Reviewer for his/her careful check of our manuscript. We apologize for the mistake in referencing.

We have also corrected the unit in 1um as suggested.

We again thank you the Reviewer.
